# Improving micromorphological analysis with CNN-based segmentation of flint/obsidian, bone and charcoal

Rafael Arnay[1], Pedro García-Villa[2], Javier Hernández-Aceituno[1]*,
Sara Rueda-Saiz[2], Carolina Mallol[2]

1 Department of Computer Engineering and Systems, University of La Laguna, La Laguna, Santa Cruz de Tenerife, Spain, 2 Archaeological Micromorphology and Biomarker Research Lab, Instituto Universitario de Bio-Orgánica Antonio González (IUBO), La Laguna, Santa Cruz de Tenerife, Spain

* jhernaac@ull.es

## Abstract

The quantification and identification of components in archaeological micromorphology remain subjective and challenging, particularly for early-career researchers. To address this, we developed a deep learning tool for the automatic segmentation of three materials commonly found in Palaeolithic contexts and thin sections: bone, charcoal, and lithic fine-grained debitage (flint and obsidian). Using high-resolution photomicrographs of 57 thin sections in plane-polarised and cross-polarised light, we trained and evaluated state-of-the-art convolutional neural networks (CNNs) for material segmentation. The best-performing configuration, a U-Net with an InceptionV4 encoder, achieved mean intersection over union (IoU) scores of 0.96 for flint/obsidian, 0.80 for bone, and 0.82 for charcoal. The models also classified the relative abundance of each material with balanced accuracies of 0.99 for flint/obsidian, 0.92 for bone, and 0.85 for charcoal. These results demonstrate the potential of deep learning to enhance objectivity, accuracy, and reproducibility in archaeological micromorphology, providing a valuable resource for future geoarchaeological research.

## Introduction

Micromorphology, particularly archaeological micromorphology, where natural and anthropogenic components co-exist, always retains a certain degree of subjectivity. The identification of these components is often challenging, especially for early-career researchers or those without a strong background in mineralogy and soil sciences [1]. Additionally, quantifying the abundance of components in thin section introduces further complexity and subjectivity. Most micromorphologists follow the foundational guidelines established by Bullock et al. (1985) [2], later revised and expanded by Stoops [3]. These general guidelines include a brief section on quantifying component abundance, which has become widely adopted in the field ([3], Fig. 4.12). This method relies on visual estimation using the microscope's field of

**Data availability statement:** All code created as part of the presented work and the test data used in the experiments are available at the following public repository "CNN-flint-bone-char", located at https://github.com/jhaceituno/CNN-flint-bone-char.

**Funding:** The presented work was funded by project "PID2023-150177NB-100 MPHEARTHS", granted by the "Agencia Estatal de Investigaci'on" of the Government of Spain to co-author Carolina Mallol. The funders had no role in study design, data collection and analysis, decision to publish, or preparation of the manuscript.

**Competing interests:** The authors have declared that no competing interests exist.

view to assess the surface area occupied by each component. Based on this, abundance is categorized as Very Few (<5%), Few (6-15%), Common (16-30%), Frequent (31-50%), Dominant (51-70%) and Very Dominant (>70%). An alternative method, although less commonly used, involves simplified scales that classify abundance as "low", "medium" or "high", often represented numerically (e.g., 1. 2. 3) or symbolically (e.g., "+" for low abundance and "+++" for high abundance) [4,5].

These classification systems, however, are not without issues. As Stoops ([3], p. 57) notes, they are inherently subjective: "The abundancy of fabric units can be expressed in different ways, such as the number of units per surface unit (e.g. the number of quartz grains per square centimeter), or the surface percentage occupied by a partial fabric". A good example of this ambiguity is the quantification of pedofeatures and porosity features.

For instance, in a thin section dominated by channels, all of which possess limpid clay coatings, it becomes difficult to isolate and measure the coated surfaces precisely. According to Stoops' framework, the coatings might be classified as Very Few (<5%). However, since most or all of the porosity features are coated, one could argue for a Dominant (51-70%) or even Very Dominant (>70%) classification when considering relative abundance within porosity features.

Given these challenges, some researchers opt to quantify only one or two main components of the sequence, or choose not to quantify abundance at all. Instead, they describe the overall composition of the sample or stratigraphic unit and highlight its key features [6–13]. Others do include quantitative abundance data, either in the main text or in the appendixes [14–20].

Archaeological and geoarchaeological methods are currently advancing at a high pace [21], with the rapid integration of technological and computational advances across all research areas. This evolution is facilitating the development of innovative methodologies as well as the refinement of established ones. It is plausible that, in the near future, much of the data processing in these disciplines will be carried out through machine learning (ML) techniques.

As outlined in the literature, ML and image recognition are beginning to be applied in all observational scales: macro, meso, and micro. At the macro-scale, applications include the detection of archaeological sites [22], predictive modelling for site location [23], resource distribution modelling [24], and site chronology prediction [25]. At the meso-scale, image-based ML has been used to compare manually and computationally conducted traceological analyses [26], often demonstrating high reliability and even increased accuracy in some cases [27]. ML has also been applied in zooarchaeology for the identification and classification of faunal remains [28]. At the micro-scale, ML has been employed in use-wear analysis [29] and micromorphology, for instance, in the identification of porosity features [30], mineral detection [31] and general thin section analysis [32].

To enhance the reliability of micromorphological data, it is essential to maximize objectivity and, where possible, introduce quantitative precision to support interpretation. With this in mind, the primary aim of our project is to develop a tool that uses image-based ML to facilitate the identification and quantification of some of the most

common materials in Palaeolithic contexts and thin sections: bone (fresh, degraded, burnt and calcined), charcoal and fine-grained lithic debitage (flint and obsidian).

## Materials and methods

This study employs deep learning techniques for semantic segmentation of soil photomicrographs, aiming to detect and differentiate three archaeological components: lithic fine-grained debitage (flint and obsidian), charcoal, and bone remains. Due to an insufficient sample size of flint, we decided to include obsidian fragments in the analysis. Although these materials are different mineralogically and can be distinguished using Crossed Polarized Light (XPL), their conchoideal type of fracture, their shape, size, and optical characteristics under Plane Polarized Light (PPL) are similar. For the sake of consistency, the label *flint* was retained, although the material under analysis more accurately corresponds to fine-grained lithic debitage. These photomicrographs were obtained from high-resolution scans of 57 micromorphological thin sections from both archaeological and experimental contexts where these materials are abundant. Chronologies and geographical areas include European and Near Eastern Middle Palaeolithic, Iberian Peninsula Neolithic, Ancient Canarian and experimental samples studied in [33].

Of these, 51 were taken under PPL and 6 under XPL. XPL scans were used to address different colours of flint and bone in PPL and XPL. All of the analyzed thin sections are housed at the Archaeological Micromorphology and Biomarkers Research Laboratory (AMBILAB), University of La Laguna, Tenerife. No permits were required for the described study, which complied with all relevant regulations.

## Dataset

The dataset is obtained from labeled high-resolution photomicrographs. All input images were segmented and annotated with the *LabelMe* tool [34], identifying regions corresponding to three structural types: flint/obsidian, bone, charcoal, and background (used when none of the other categories are present). 51 of the thin sections were annotated directly by authors García-Villa and Rueda-Saiz and 6 were annotated by trainees and interns L.M, A.T., A.N and I.M. They were trained and supervised by P.G-V and C.M. and were prepared beforehand with the basics of micromorphology, optical petrography and archaeological remains recognition in thin section. Quality control was assured through assisted annotation and random spot checks.

Both the original images and their corresponding label masks are subdivided into 2000x2000 pixels images. The sub-images that contain at least one pixel corresponding to a given class are included in the dataset for that class. Fig 1 shows some examples of images and their corresponding segmentation masks for each class. In total, there are 820 images for flint/obsidian, 2519 for bone, and 1715 for charcoal. The data are randomly split into training (70%), validation (20%), and test (10%) subsets while ensuring no overlap between the sets. Table 1 shows the number of training, validation and test images per class.

Each image and mask are resized to a resolution of 512×512 pixels. To increase the diversity of training data, random horizontal and vertical flips are applied as data augmentation techniques using the `Albumentations` library [35]. Table 2 shows the relative size of regions compared to the image size and number of regions per image for each dataset.

## Model architectures and encoders

Modern deep learning-based image segmentation models typically adopt an encoder–decoder architecture, in which the encoder extracts hierarchical features from the input image, and the decoder reconstructs the segmentation map from these features. In this work, we adopt a modular framework where different segmentation heads (e.g., U-Net [36], FPN [37], PSPNet [38]) are combined with various pre-trained convolutional encoders (ResNet [39], VGG [40], Inception [41], Xception [42]).

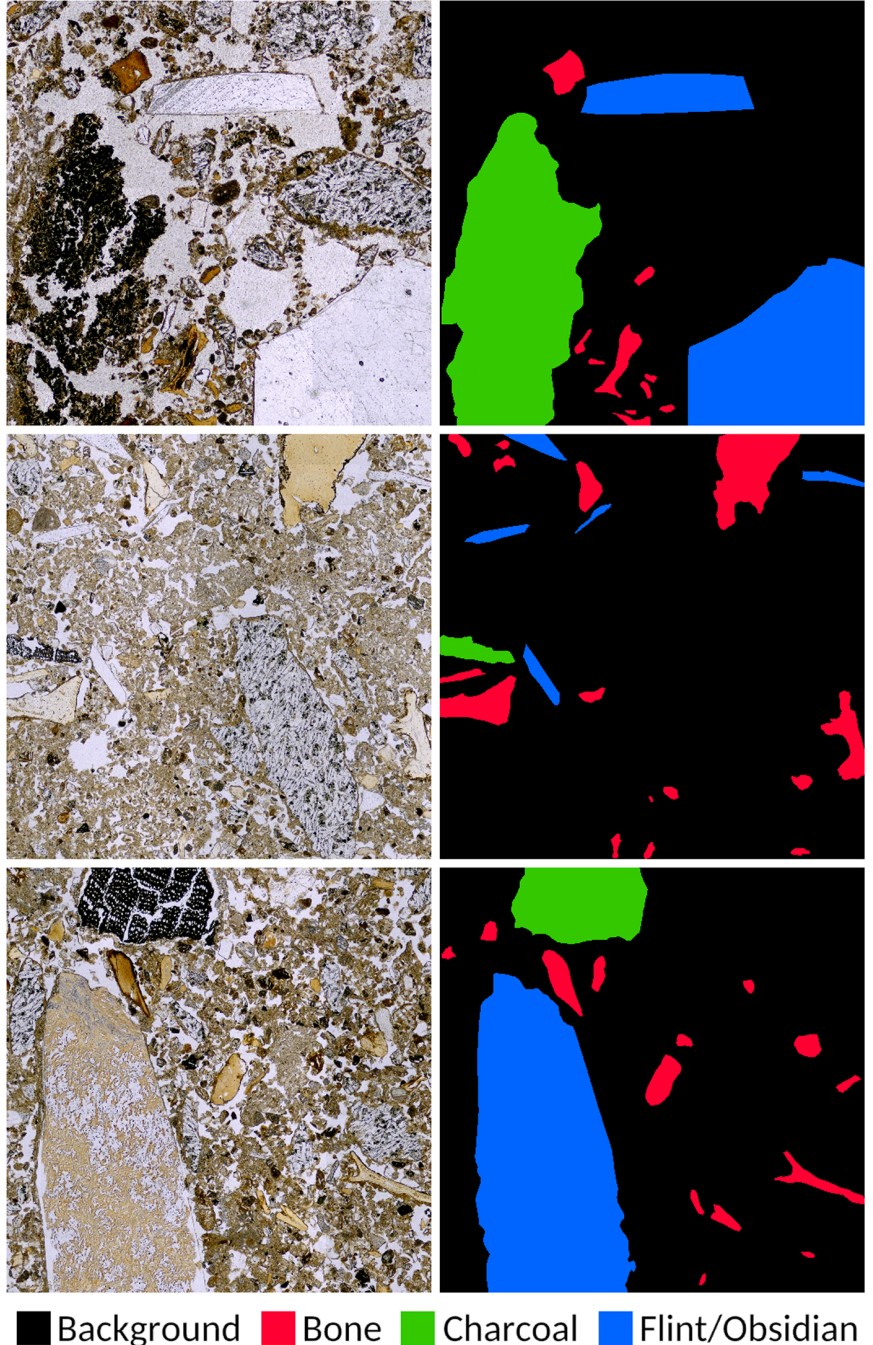

**Fig 1**. Photomicrographs (left) and labels (right).

**Table 1**. Number of images in the train, validation and test sets for each class.

| Class | Train | Validation | Test |
|---|---|---|---|
| Flint/Obsidian | 574 | 164 | 82 |
| Bone | 1763 | 503 | 253 |
| Charcoal | 1200 | 343 | 172 |

**Table 2**. Relative size of regions compared to the image size and number of regions per image.

| | Relative size | | Number of regions | |
|---|---|---|---|---|
| Class | mean | std | mean | std |
| Flint/Obsidian | 0.02 | 0.16 | 9.35 | 9.45 |
| Bone | 0.01 | 0.04 | 13.87 | 24.76 |
| Charcoal | 0.01 | 0.05 | 16.49 | 20.75 |

**Encoders.** The encoder, or backbone, is usually derived from a classification network trained on large-scale datasets such as ImageNet. These encoders consist of multiple stages of convolutional and pooling layers that progressively reduce spatial resolution while increasing feature abstraction. To use them for segmentation, the classification head is removed, and the feature maps from different encoder stages are routed to the decoder.

**Decoders.** The decoder reconstructs a dense segmentation map from the hierarchical features extracted by the encoder. Each architecture handles this differently:

- **U-Net** employs a symmetric structure with skip connections between encoder and decoder layers of equal resolution [36].
- **FPN** combines features from different depths in a top-down fashion to handle multi-scale representations [37].
- **PSPNet** uses pyramid pooling to incorporate global context at multiple receptive fields [38].

**Integration procedure.** The combination of encoder and decoder components proceeds as follows:

1. **Backbone selection:** Choose an encoder network and remove its fully connected layers.
2. **Pre-training:** Load ImageNet pre-trained weights to leverage transfer learning.
3. **Feature tapping:** Extract intermediate feature maps from key layers in the encoder.
4. **Decoder setup:** Assemble the decoder to progressively upsample and fuse the feature maps to the original image resolution.
5. **Output head:** Apply a final convolutional layer to produce logits in order to calculate the Dice loss.

## Loss function and optimisation

A Dice Loss function was used to optimise model performance. The Dice loss is a commonly used metric in semantic segmentation tasks to evaluate the overlap between predicted and ground truth regions [43]. It is derived from the Dice Similarity Coefficient (DSC), and is particularly effective for handling imbalanced class distributions.

Given the predicted segmentation mask $P$ and the ground truth mask $G$, both represented as binary vectors, the Dice coefficient is defined as:

$$\mathrm{DSC}(P, G) = \frac{2 \sum_i p_i g_i}{\sum_i p_i + \sum_i g_i} \qquad (1)$$

where $p_i$ and $g_i$ are the predicted and ground truth values for pixel $i$, respectively. The Dice loss is then defined as:

$$\mathcal{L}_{\mathrm{Dice}} = 1 - \frac{2 \sum_i p_i g_i + \epsilon}{\sum_i p_i + \sum_i g_i + \epsilon} \qquad (2)$$

Here, $\epsilon$ is a small constant added to avoid division by zero. The Dice loss ranges from 0 (perfect overlap) to 1 (no overlap), and is minimized during training to improve segmentation accuracy. The Adam optimizer [44] was selected with a learning rate of $1 \times 10^{-4}$. All models were trained for a maximum of 40 epochs with a batch size of 16.

## Model training and evaluation

The training loop monitors the validation Intersection-over-Union (IoU) [45] metric and saves the best-performing model checkpoint for evaluation on the Test set. The IoU, also known as the Jaccard Index, is a standard evaluation metric for measuring the overlap between two regions, typically a predicted segmentation mask $P$ and the ground truth mask $G$. It is defined as the ratio between the area of intersection and the area of union of the predicted and ground truth regions:

$$\text{IoU}(P, G) = \frac{|P \cap G|}{|P \cup G|} = \frac{\sum_i p_i g_i}{\sum_i p_i + \sum_i g_i - \sum_i p_i g_i} \tag{3}$$

where $p_i$ and $g_i$ represent the binary values (0 or 1) of the predicted and ground truth masks at pixel $i$, respectively. The IoU metric ranges from 0 to 1. where a value of 1 indicates perfect overlap, and a value of 0 indicates no overlap. It is widely used in segmentation and object detection benchmarks due to its ability to penalise both false positives and false negatives. Two evaluation strategies were employed:

- **Per-image IoU:** Calculates IoU individually for each image and reports the average.
- **Dataset-level IoU:** Aggregates pixel-level true positives, false positives, and false negatives over the entire dataset before computing IoU.

Per-image IoU provides insight into the model's consistency across individual samples, it is more sensitive to local errors, particularly in images with small or sparse target regions. In microscopic archaeological images, these regions – such as tiny fragments of bone, flint/obsidian, or charcoal – may occupy only a minimal portion of the image. As a result, even minor segmentation inaccuracies can cause significant drops in per-image IoU. Moreover, in cases where the target class is nearly absent or fragmented, the metric may be disproportionately penalized, despite acceptable qualitative performance. Conversely, dataset-level IoU tends to smooth out such sample-level fluctuations by emphasizing the overall class distribution and model behaviour across the entire dataset. This makes it a more stable metric for comparative analysis across models, especially when dealing with class imbalance and small object segmentation.

In addition to assessing the system's performance on the segmentation task, we also aimed to evaluate the system on the classification task, as proposed by [3]. Specifically, this involved classifying the images based on the relative abundance of a given component (bone, charcoal, or flint/obsidian) into five categories: Very Few (<5%), Few (6–15%), Common (16–30%), Frequent (31–50%), Dominant (51–70%), and Very Dominant (>70%). Fig 2 shows a workflow diagram of the segmentation and classification process.

## Results

Tables 3, 4 and 5 summarise the performance of the different segmentation models across target classes using various combinations of architectures and encoders. Evaluation metrics include IoU computed per image and across the entire dataset, for the test sets.

Among all configurations, the highest dataset-level IoU was achieved by the U-Net architecture using the InceptionV4 encoder for the flint/obsidian class, reaching an IoU of 0.96. This indicates exceptional segmentation accuracy for identifying flint/obsidian particles in microscopic soil images. FPN combined with Xception also yielded competitive results for the same class (IoU = 0.95).

In general, models targeting flint/obsidian outperformed those aimed at detecting charcoal and bone. The best bone segmentation model (Unet + InceptionV4) reached an IoU of 0.80, while charcoal segmentation peaked at 0.82 with U-Net + InceptionV4 and Xception. Figs 3, 4 and 5 show a qualitative comparison between the ground truth segmentations and those obtained using the Unet + Inceptionv4 model, for bone, charcoal and flint/obsidian, respectively.

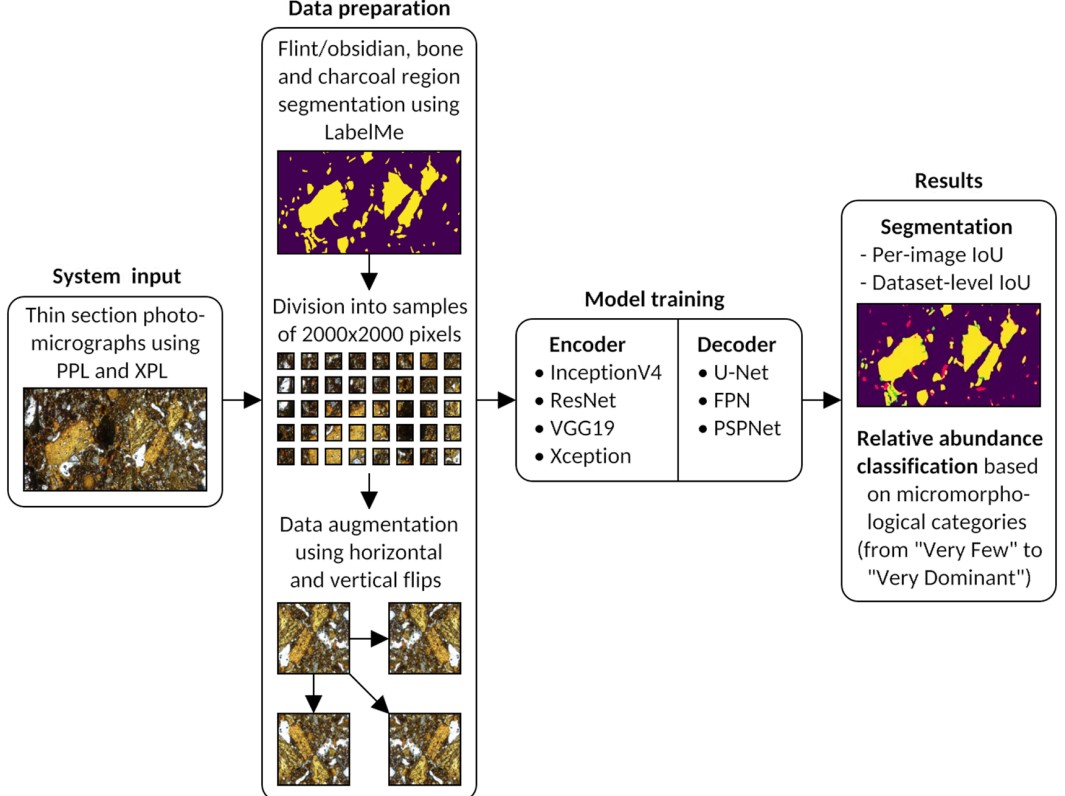

**Fig 2. Segmentation and classification workflow.**

**Table 3. Model performance in the bone class.**

| Architecture | Encoder | per image IoU | dataset IoU |
|---|---|---|---|
| FPN | inceptionv4 | 0.60 | 0.77 |
| | resnet50 | 0.58 | 0.75 |
| | vgg19 | 0.62 | 0.74 |
| | xception | 0.55 | 0.75 |
| PSPNet | inceptionv4 | 0.52 | 0.72 |
| | resnet50 | 0.49 | 0.71 |
| | vgg19 | 0.56 | 0.74 |
| | xception | 0.45 | 0.65 |
| Unet | inceptionv4 | 0.61 | 0.80 |
| | resnet50 | 0.60 | 0.78 |
| | vgg19 | 0.62 | 0.76 |
| | xception | 0.61 | 0.70 |

Performance of the different segmentation models in the bone class using various combinations of architectures and encoders.

The distribution of abundance categories as described in [3] within the test sets for flint/obsidian, bone, and charcoal is presented in Table 6. This distribution is notably unbalanced across the three structures: for all materials, the "Very Few" category dominates the dataset, particularly for bone (79.05%) and charcoal (64.53%), while flint/obsidian shows a slightly more balanced distribution, though still skewed towards "Very Few" (53.66%). Higher abundance categories such as

**Table 4. Model performance in the charcoal class.**

| Architecture | Encoder | per image IoU | dataset IoU |
|---|---|---|---|
| FPN | inceptionv4 | 0.59 | 0.81 |
| | resnet50 | 0.59 | 0.81 |
| | vgg19 | 0.57 | 0.80 |
| | xception | 0.57 | 0.81 |
| PSPNet | inceptionv4 | 0.49 | 0.78 |
| | resnet50 | 0.47 | 0.78 |
| | vgg19 | 0.53 | 0.80 |
| | xception | 0.46 | 0.75 |
| Unet | inceptionv4 | 0.59 | 0.82 |
| | resnet50 | 0.62 | 0.81 |
| | vgg19 | 0.59 | 0.80 |
| | xception | 0.60 | 0.82 |

Performance of the different segmentation models in the charcoal class using various combinations of architectures and encoders.

**Table 5. Model performance in the flint/obsidian class.**

| Architecture | Encoder | per image IoU | dataset IoU |
|---|---|---|---|
| FPN | inceptionv4 | 0.64 | 0.93 |
| | resnet50 | 0.67 | 0.92 |
| | vgg19 | 0.68 | 0.91 |
| | xception | 0.64 | 0.95 |
| PSPNet | inceptionv4 | 0.60 | 0.92 |
| | resnet50 | 0.61 | 0.89 |
| | vgg19 | 0.66 | 0.90 |
| | xception | 0.45 | 0.85 |
| Unet | inceptionv4 | 0.69 | 0.96 |
| | resnet50 | 0.67 | 0.95 |
| | vgg19 | 0.64 | 0.90 |
| | xception | 0.71 | 0.94 |

Performance of the different segmentation models in the flint/obsidian class using various combinations of architectures and encoders.

"Frequent", "Dominant", and "Very Dominant" are underrepresented in all three materials, which may pose challenges for the classifier when predicting these less frequent classes.

Despite this class imbalance, the classification results using the U-Net model with InceptionV4 as encoder and summarized in Table 7 indicate that the system performs remarkably well across all materials. For flint/obsidian, the model achieves near-perfect performance with a balanced accuracy, precision, recall, and F1-score of 0.99. Bone also demonstrates high classification performance, with a balanced accuracy of 0.92 and an F1-score of 0.94. showing that the model can handle the dominant "Very Few" class as well as the less represented ones. Charcoal presents the lowest performance among the three materials, with a balanced accuracy and F1-score around 0.85 (Fig 6).

## Discussion

### Class performance and methodological considerations

The performance of the models varied substantially across classes, which may reflect intrinsic differences in the visual distinctiveness and morphological consistency of each material class. U-Net generally delivered more robust results than the other architectures. This could be attributed to U-Net's skip connections, which facilitate precise localisation by preserving spatial resolution, a critical factor when dealing with fine-grained structures like micro-remains.

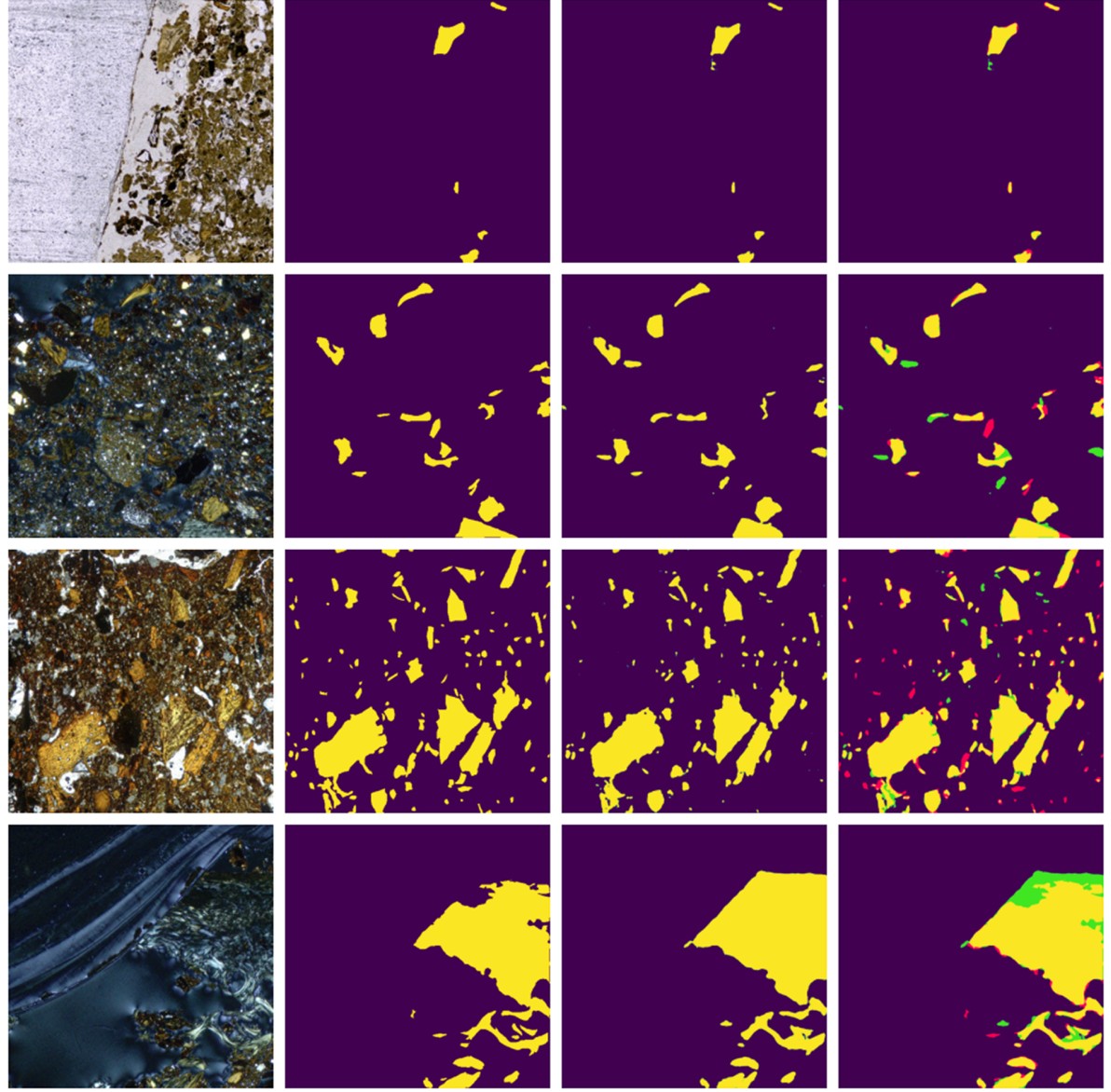

**Fig 3**. Bone class photomicrograph, ground truth, prediction and comparison of the segmentation, where false positives are shown in green and false negatives in red.

The variation in encoder performance also reveals interesting trends. InceptionV4 consistently performed well across classes, potentially due to its multi-scale feature extraction capabilities, which are beneficial when segmenting complex textures found in archaeological samples. Interestingly, the dataset-level IoUs were consistently higher than the per-image values, which suggests that even though some individual predictions may underperform, the models perform robustly across the dataset as a whole.

Flint/obsidian consistently achieved the highest accuracy (IoU up to 0.96), reflecting its strong optical distinctiveness under PPL and relatively uniform morphological features. In contrast, bone and charcoal yielded lower IoUs (0.80 and 0.82. respectively). These differences highlight that segmentation quality depends not only on network architecture but

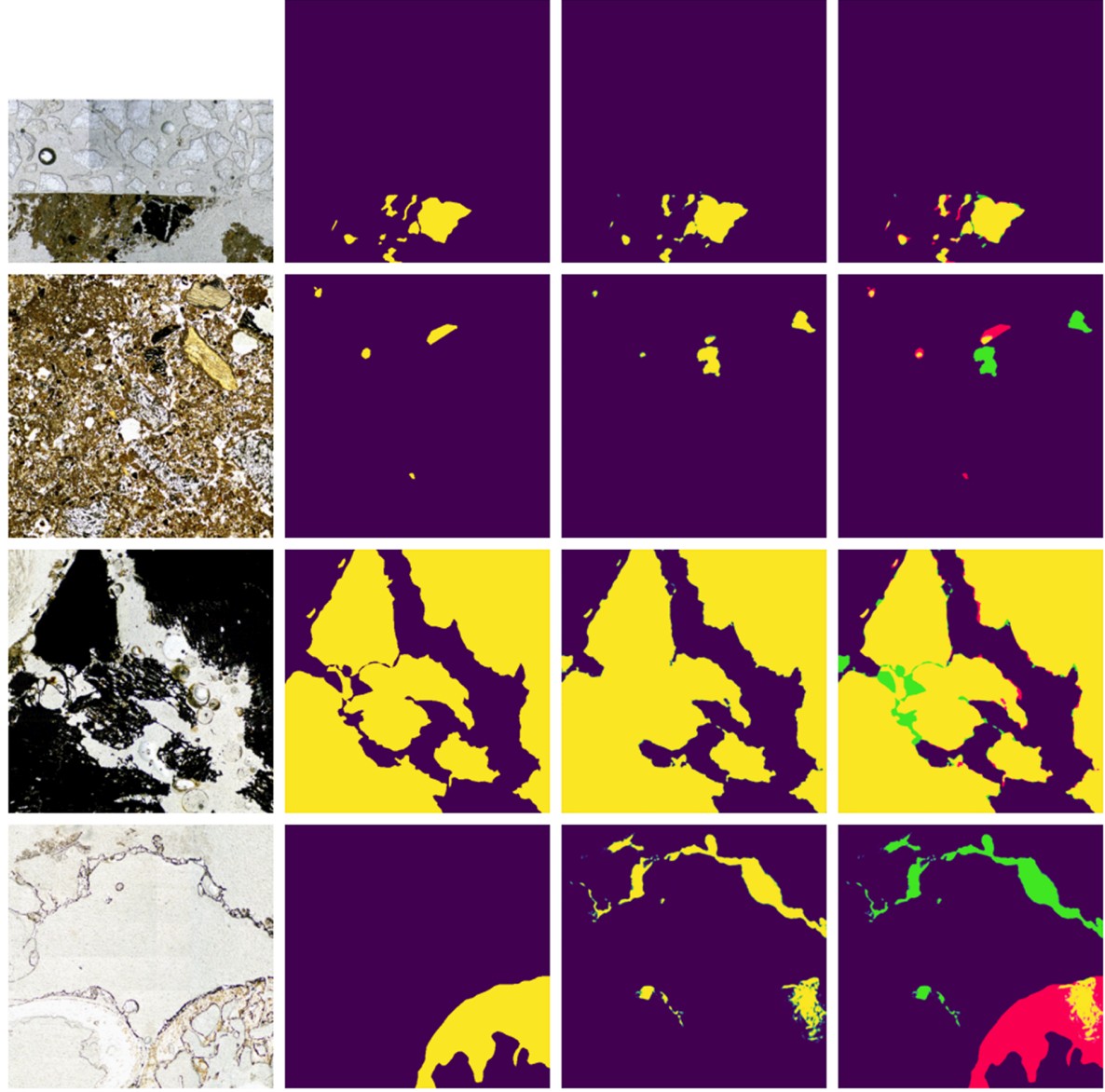

**Fig 4**. Charcoal class photomicrograph, ground truth, prediction and comparison of the segmentation, where false positives are shown in green and false negatives in red.

also on the inherent variability of each material. It may also reflect the increased difficulty in classifying charcoal, potentially due to more subtle abundance differences or more complex image features.

Bone fragments exhibit heterogeneous textures, colours and shapes, as their preservation state varies from fresh to degraded or burnt, while charcoal often displays a wide range of inner pore variability, based on its species and orientation cut. Such different properties, as well as the annotation difficulties faced (see "Annotating difficulties and future directions" below) likely explain the lower performance observed for these classes.

Typical failure modes further illustrate these challenges. Misclassification of bone often occurred when fragments were small, degraded, or heavily diagenetically altered, producing visual signatures close to the background and other

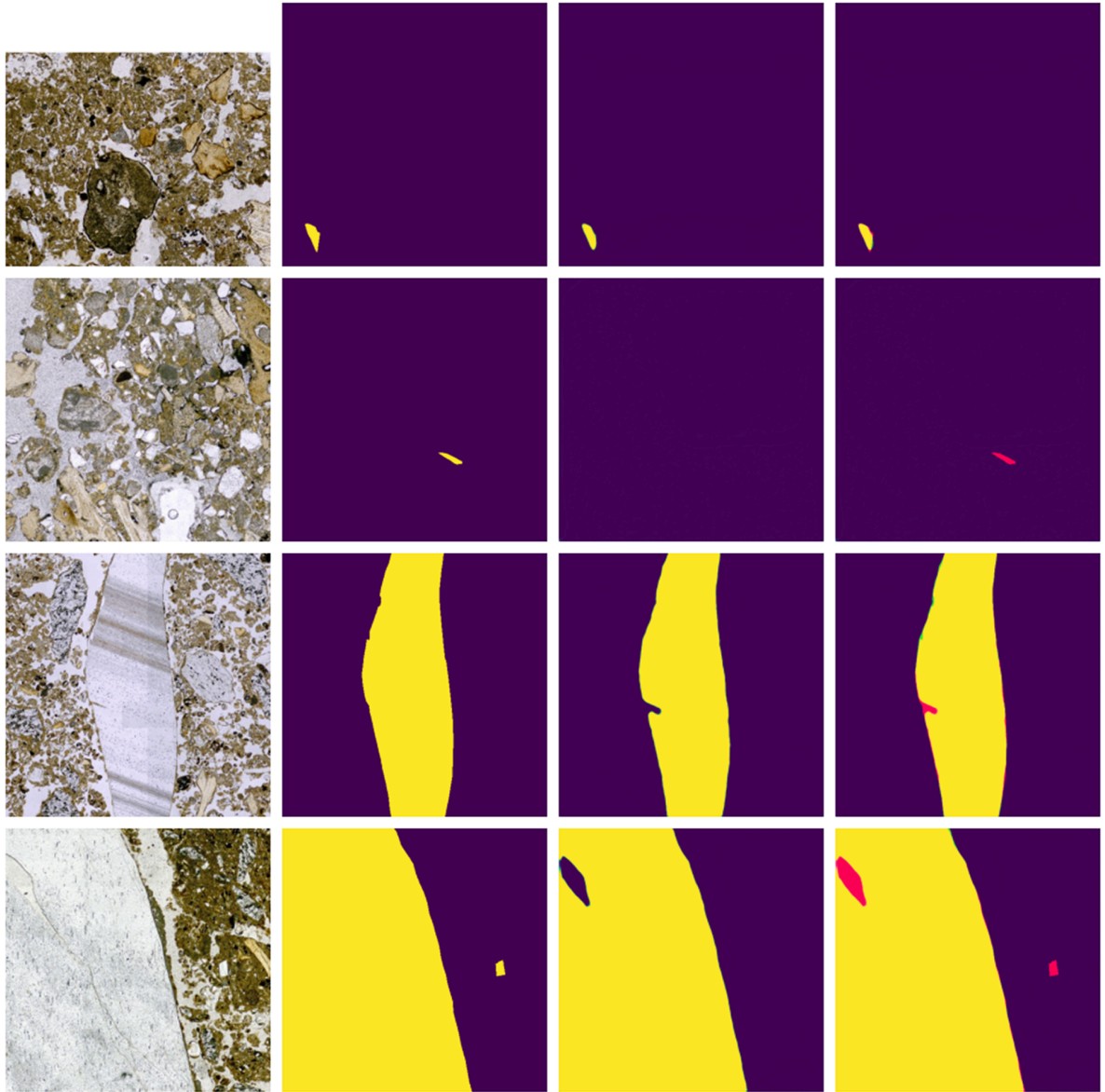

**Fig 5**. Flint/obsidian class photomicrograph, ground truth, prediction and comparison of the segmentation, where false positives are shown in green and false negatives in red.

**Table 6**. Number and percentage of images per abundance level.

|  | Very Few | Few | Common | Freq. | Dom. | V.D. |
|---|---|---|---|---|---|---|
| Flint/Obs. | 44(53.7%) | 19(23.2%) | 5(6.1%) | 3(3.7%) | 5(6.1%) | 6(7.3%) |
| Bone | 200(79.1%) | 31(12.3%) | 16(6.3%) | 5(2.0%) | 0(0%) | 1(0.4%) |
| Charcoal | 111(64.5%) | 30(17.4%) | 13(7.6%) | 8(4.7%) | 4(2.3%) | 6(3.5%) |

Number of images for each abundance level and their corresponding percentage relative to the total number of images in the test set. Obs.: Obsidian, Freq.: Frequent, Dom.: Dominant, V.D.: Very Dominant.

**Table 7**. Classification performance across abundance categories using U-Net with InceptionV4.

|  | Balanced Acc. | Precision | Recall | f1-score |
|---|---|---|---|---|
| Flint/Obsidian | 0.99 | 0.99 | 0.99 | 0.99 |
| Bone | 0.92 | 0.96 | 0.92 | 0.94 |
| Charcoal | 0.85 | 0.85 | 0.85 | 0.84 |

Classification performance for the flint/obsidian, bone, and charcoal test sets across abundance categories using the U-Net model with InceptionV4 as encoder. Metrics reported include balanced accuracy, precision (macro averaged), recall (macro averaged), and F1-score (macro averaged).

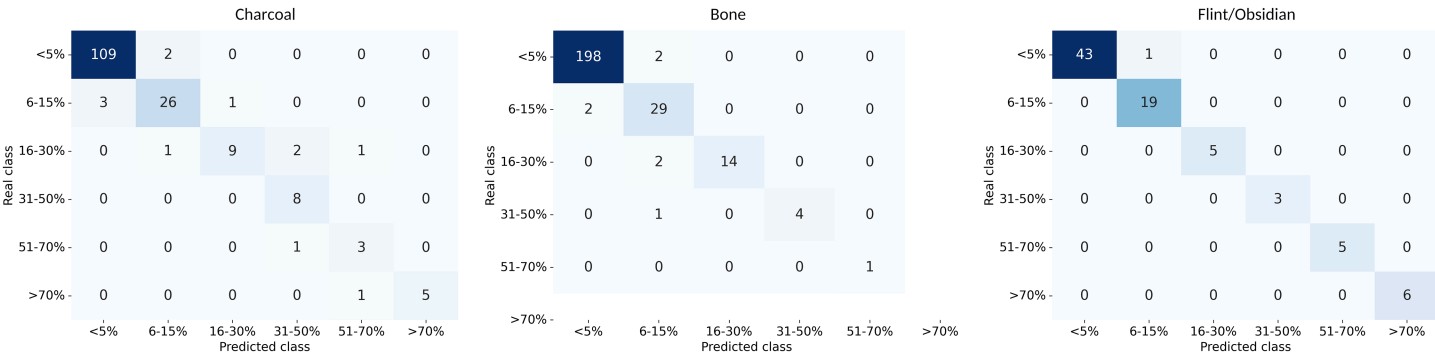

**Fig 6**. **Confusion matrices of relative abundance classification.** Confusion matrices obtained for the classification of relative abundance using the U-Net model with InceptionV4 as encoder in the test sets for flint/obsidian, charcoal and bone. The categories are: Very Few (<5%), Few (6–15%), Common (16–30%), Frequent (31–50%), Dominant (51–70%), and Very Dominant (>70%).

archaeological components. Charcoal errors were frequent in thin or faint fragments, which the model occasionally ignored altogether, biasing predictions toward the dominant "Very Few" category.

Annotation quality also influenced outcomes. All segmentations were produced manually, and even minor inconsistencies in defining boundaries or labeling fine fragments can propagate into training. This effect is compounded by the strong class imbalance in the dataset. Most samples fell into the "Very Few" abundance category, while higher abundance levels ("Frequent", "Dominant", "Very Dominant") were underrepresented. As a result, the models were well-calibrated for sparse occurrences but they may be less reliable when confronted with denser distributions. However, as evidenced by confusion matrices the models are able to predict the most dominant classes quite well.

These performance dynamics have direct implications for applying Stoops' abundance standards. Automated classifications of "Very Few" or "Few" appear robust for flint/obsidian, but bone and charcoal predictions are less reliable in higher-abundance classes, where underestimation may systematically bias interpretation. In practice, this means that while the system could assist micromorphologists in consistently flagging low-abundance categories, manual verification remains essential when assessing more common or dominant occurrences. For field and laboratory decision-making, this distinction is crucial: overconfidence in model outputs for bone and charcoal could distort reconstructions of combustion features, depositional processes, or occupation intensity.

Finally, there is a methodological risk of data leakage. Although images were randomly split into training, validation, and test sets, thin-section subimages derived from the same slide may share local structures or textures. If such near-duplicates occur across splits, performance estimates could be inflated.

## Archaeological impact and applicability

Charcoal, bone, and fine-grained lithic remains are among the most common materials found in thin sections across Palaeolithic contexts. The performance of the model indicates that its application is not restricted to the Palaeolithic but

can be extended to any setting where these materials occur. This makes the approach broadly transferable and highly reproducible, since all that is required is a high-resolution scan of the thin section.

This methodology would enhance the quantification of these elements in archaeological features. For instance, in the characterization of combustion structures, where charcoals are frequently found, or the identification of surface occupations and activity areas, characterized by high abundance of bones and fine-grained lithic debitage.

Nevertheless, some challenges remain. Identification might be more difficult when materials have undergone strong taphonomic alteration, causing bones and charcoal to lose their typical colour and morphology under PPL. Likewise, fine-grained debitage can vary in colour depending on raw material, and coarse lithic artefacts without a pronounced conchoidal fracture are difficult to distinguish – both for the CNN and for specialists – from natural rock fragments.

Even so, the archaeological significance of the results is clear. Minor discrepancies in IoU values do not imply that fragments are missed, but rather that the CNN's boundaries do not align perfectly with the annotated polygons. For example, the lowest-performing case – charcoal identified with the U-Net model and InceptionV4 encoder – still achieved an IoU of 0.84.

Taken together, these outcomes demonstrate the potential impact of the method on archaeological practice. The possibility of automating part of the identification and quantification process constitutes a practical improvement: it would reduce the time micromorphologists spend on manual annotation and counting, while still providing results that are consistent and dependable.

### Annotating difficulties and future directions

Human error during the annotation stage proved to be a decisive factor in the final recognition of elements. Several difficulties were identified that should guide future improvements.

*LabelMe*, although a practical tool, is not the best suited for this task. It only allows polygons built from straight lines, while most bone and charcoal fragments have rounded or curved edges. This mismatch forced the annotation of highly rounded objects to be done with hundreds of tiny segments, making the process slow and often imprecise. By contrast, flint and obsidian – with their sharp edges and angular forms – were far easier to annotate, requiring far fewer steps to achieve a close fit. Very small particles ($< 50\,\mu m$) posed another problem: annotating them demanded such high magnification that overall resolution was lost, producing irregular and uncertain boundaries.

Addressing these issues will be central in the next stage of research. The annotation tool itself should be replaced or upgraded with software that handles curved outlines more efficiently, easing the annotation of rounded fragments that dominate the archaeological record. At the same time, the dataset will be expanded to include a greater number of these materials, along with additional fine-grained lithic raw materials such as chert and quartzite, as well as charcoal exhibiting greater morphological variability and bone fragments. Other materials expected to be incorporated into the dataset include pottery fragments, specific rock lithologies, and malacofauna, thereby broadening the applicability of the model.

From a micromorphological standpoint, the next step is to reassemble individual images into full thin sections, so that results can be delivered not only at the scale of the section as a whole, but also at finer levels of analysis—such as defined microfacies or combustion features. These adjustments will improve both the accuracy of annotations and the archaeological relevance of the outputs.

### Conclusion

This study presents a successful application of deep learning techniques for the semantic segmentation and abundance classification of archaeological micro-remains in soil thin sections. The U-Net model, particularly when combined with the InceptionV4 encoder, consistently outperformed other configurations across all target materials, achieving high segmentation accuracy, especially for flint/obsidian. The results validate the feasibility of automating material

identification in micromorphological images, reducing subjectivity and increasing reproducibility in archaeological analysis. The developed system also proved capable of accurately classifying the relative abundance of materials, even under class imbalance conditions. Although flint/obsidian achieved the highest segmentation and classification performance, bone and charcoal also yielded promising results, suggesting that this approach can be generalized to other microremains in archaeological contexts. Future work should focus on expanding the dataset, especially in underrepresented abundance categories, and exploring the application of this methodology to additional materials and archaeological sites.

This study demonstrates the potential of deep learning for segmenting and classifying key archaeological microremains—flint/obsidian, bone, and charcoal—in thin sections. The U-Net with an InceptionV4 encoder achieved high segmentation accuracy, especially for flint/obsidian, and the models were also able to categorise material abundance levels in line with Stoops' standards. These outcomes suggest that automated approaches can complement traditional micromorphological description by providing consistent quantitative measures that may inform laboratory and field decisions about site formation, combustion features, and occupation evidence.

At the same time, several limitations constrain the generalisability of these results. The dataset derives from a single laboratory collection (57 thin sections), with a strong PPL bias (51 of 57 images), and marked class imbalance—particularly the overrepresentation of "Very Few" abundance categories. The merging of flint and obsidian into a single class reflects the small sample size and raises the risk of conflating distinct materials. Furthermore, several archaeological contexts use chert, quartzite or other fine and coarse-grained materials that are not represented in this study. Split leakage between training and evaluation sets cannot be entirely ruled out. Performance also varied considerably across classes, with excellent results for flint/obsidian but weaker outcomes for bone and charcoal.

Future work should expand the dataset to include more balanced representation of abundance categories and other microremains, incorporate cross-lab validation, and report run-to-run uncertainty. Within those bounds, deep learning has promise as a support tool—rather than a replacement—for micromorphological practice, helping to operationalise Stoops' abundance classes and guide archaeological interpretation in both laboratory and field contexts.

## Acknowledgments

The authors wish to thank L.M for their voluntary participation in tagging the microphotographs, and A.T., A.N and I.M. for their participation in the project within the framework of their bachelor's internship.

## Author contributions

**Conceptualization:** Rafael Arnay, Pedro García-Villa, Carolina Mallol.

**Data curation:** Pedro García-Villa, Sara Rueda-Saiz, Carolina Mallol.

**Methodology:** Rafael Arnay.

**Software:** Rafael Arnay, Javier Hernández-Aceituno.

**Supervision:** Rafael Arnay, Pedro García-Villa.

**Validation:** Rafael Arnay.

**Visualization:** Javier Hernández-Aceituno.

**Writing – original draft:** Rafael Arnay, Pedro García-Villa, Sara Rueda-Saiz.

**Writing – review & editing:** Rafael Arnay, Pedro García-Villa, Javier Hernández-Aceituno.

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
