## [Editor Report · Decision Letter 0]

20 Aug 2025

PONE-D-25-43021Improving micromorphological analysis with CNN-based segmentation of flint/obsidian, bone and charcoalPLOS ONE

Dear Dr. Hernández-Aceituno,

Thank you for submitting your manuscript to PLOS ONE. After careful consideration, we feel that it has merit but does not fully meet PLOS ONE’s publication criteria as it currently stands. Therefore, we invite you to submit a revised version of the manuscript that addresses the points raised during the review process.

Major Revision, grounded in PLOS ONE’s criteria of methodological soundness, transparency and reproducibility rather than novelty; specific content-level comments follow below.

For acceptance, the authors must provide repository DOIs for a minimal, reproducible dataset and all author-generated code, and restructure the manuscript by separating Results from Discussion and tempering the Conclusions; the remaining language, formatting and minor presentation edits are recommended to improve clarity.

We look forward to receiving your revised manuscript.

Kind regards,

Przemysław Mroczek, Dr. hab.

Academic Editor

PLOS ONE

Journal Requirements:

2. In your manuscript, please provide additional information regarding the specimens used in your study. Ensure that you have reported human remain specimen numbers and complete repository information, including museum name and geographic location.

For more information on PLOS One's requirements for paleontology and archeology research, see https://journals.plos.org/plosone/s/submission-guidelines#loc-paleontology-and-archaeology-research.

3. Please note that PLOS One has specific guidelines on code sharing for submissions in which author-generated code underpins the findings in the manuscript. In these cases, we expect all author-generated code to be made available without restrictions upon publication of the work. Please review our guidelines at https://journals.plos.org/plosone/s/materials-and-software-sharing#loc-sharing-code and ensure that your code is shared in a way that follows best practice and facilitates reproducibility and reuse.

**Additional Editor Comments:**

Having read the manuscript carefully—and as archaeological micromorphology is close to my area of expertise—I consider that the submission requires both substantive and technical revisions before it can proceed to external review. It is difficult to recruit a specialist reviewer for a paper that visibly needs structural and policy-aligned polishing; therefore, I am listing here only the content-level points that must be addressed first.

Title. Appropriate and broadly aligned with the study’s scope; no change required at this stage.

Abstract. Needs tightening and rebalancing. Please use British spelling (Palaeolithic), expand abbreviations at first mention (e.g., intersection over union (IoU)), and avoid overloading the abstract with architecture names—retain a generic “state-of-the-art CNNs” and, if essential, the single best configuration. The abstract currently foregrounds only the top metric (IoU = 0.96 for flint/obsidian); include representative values for bone and charcoal to avoid selective emphasis. Add minimal dataset context (e.g., number of thin sections, imaging modes) and provide at least one numeric value to substantiate the claim of “high balanced accuracy” for abundance classification. Stylistically, “flint (including obsidian)” reads more clearly than “flint/obsidian”.

Keywords. Streamline for discoverability. Avoid duplicating title phrases (e.g., “CNN-based”, “semantic segmentation”, “archaeological micromorphology”), avoid acronyms, standardise to UK spelling and lower case for common nouns, and remove slash forms. Keep a concise set focused on the paper’s domain/material scope without repeating title terms.

Structure (Results vs Discussion). Please separate Results and Discussion into distinct, substantive sections. The current combined section mainly reports metrics with limited interpretation, which blurs what was found versus what those findings mean. Results should be strictly descriptive (dataset composition, training/evaluation set-up, per-class/per-image performance, ablations, error rates). A standalone Discussion should interpret performance differences between classes, typical failure modes and their causes, the influence of annotation quality and class imbalance, risks of data leakage, and the extent of generalisability; situate outcomes within the literature, state limitations and potential biases, and outline practical implications for archaeological micromorphology and directions for future work.

Conclusion. Serviceable as a brief recap but currently too general and overly optimistic relative to the dataset’s constraints. The headline claim that the approach “reduces subjectivity and increases reproducibility” is not demonstrated: code and data are not yet public and validation is narrow. The Conclusion should explicitly state limitations and the bounds of generalisability (single-lab, 57 thin sections, strong PPL bias, pronounced class imbalance), and qualify any extension to “other microremains”. Assertions about reproducibility should be conditional on releasing the minimal dataset and code, with transparent train/validation/test splits and fixed random seeds. Please articulate the practical “so what?” (how segmentations and abundance classes map onto Stoops’ standards and support laboratory/field decisions) and acknowledge key methodological risks (merging flint with obsidian due to small sample size; potential split leakage; absence of uncertainty estimates across runs). The tone should be tempered to reflect disparate class performance—excellent for flint/obsidian, clearly lower for bone and charcoal—so that conclusions align with the evidence presented.

Selected technical/clarity items (for the next revision).

– Several tables use comma decimal separators (e.g., “0,60”) rather than the journal’s conventional full stop (e.g., “0.60”), and one caption truncation is visible (“Freqe”). Please standardise decimals and fix caption typos.

– Ensure consistent British spelling throughout (e.g., behaviour, organisation, optimisation). There are a few typographical slips (e.g., “structed” → “structured”).

– You note use of LabelMe and acknowledge volunteers/interns for tagging. Please add brief details on who annotated, any training, and quality control (e.g., spot checks, inter-annotator agreement), as these directly affect dataset validity.

– The dataset comprises scans of 57 thin sections curated at AMBILAB (University of La Laguna). Clarify ownership of the images and permissions to redistribute them. If the institution owns the images, please secure permission for public sharing; if not, detail the rights holder and the access mechanism.

Please revise the manuscript in line with the substantive points above. When resubmitting, include (i) a point-by-point response to every item, (ii) a clean manuscript and a marked version showing changes, and (iii) repository DOIs for the minimal dataset and author-generated code together with an updated, reader-facing Data Availability Statement. I will reassess the submission for external peer review under a decision of Major Revision once these materials are provided.

---

## [Author Response · Author response to Decision Letter 1]

23 Sep 2025

The authors greatly appreciate the provided suggestions, which have been addressed as described below:

- The style requirements for PLOS ONE have been applied to the manuscript, including authors information, figure citations, section title capitalisation, and file naming.

- Additional information regarding the permits required for our study has been included in lines 107-109 of the manuscript

- A code and data availability statement has been included at the end of the manuscript. Legal restrictions over the experimental dataset have been lifted and it is now freely available along with the code.

- The manuscript has been corrected to use British spelling and the abstract has been modified to expand all abbreviations, to avoid unnecessary architecture names, and to include representative values for bone and charcoal, minimal dataset context, and numeric values to substantiate the claim of high balanced accuracy for abundance classification. As recommended, ocurrences of "flint/obsidian" have been replaced with "flint (including obsidian)".

- Keywords have been modified to avoid duplicating title phrases, as well as acronyms and slash forms.

- The "Results" and "Discussion" sections have been separated into distinct, substantive sections. As suggested, the newly added "Discussion" section includes an interpretation of performance differences between classes, typical failure modes and their causes, the influence of annotation quality and class imbalance, risks of data leakage (subsection "Class performance and methodological considerations" and further extended in subsection "Annotating difficulties and future directions"); the extent of generalisability, the limitations and potential biases of the study, and outline practical implications for archaeological micromorphology and directions for future work (subsection "Archaeological impact and applicability").

- The conclusion section has been expanded to explicitly state the limitations and the bounds of generalisability of the presented study. As explained in the newly added "Data availabilty statement", both the code and all experimental data are now publicly available. Key methodological risks have been acknowledged and the tone of the text has been adjusted to align with the evidence presented.

- Incorrect decimal separators and table captions have been fixed.

- To the best of our knowledge, all inconsistent spelling ("optimise", "penalise", "summarise"...) and typographical mistakes have been corrected.

- Specific information regarding image tagging duties has been added to the "Dataset" subsection of "Materials and methods".

---

## [Decision Letter · Decision Letter 1]

18 Nov 2025

PONE-D-25-43021R1Improving micromorphological analysis with CNN-based segmentation of flint/obsidian, bone and charcoalPLOS ONE

Dear Dr. Hernández-Aceituno,

Thank you for submitting your manuscript to PLOS ONE. After careful consideration, we feel that it has merit but does not fully meet PLOS ONE’s publication criteria as it currently stands. Therefore, we invite you to submit a revised version of the manuscript that addresses the points raised during the review process.

The single external review obtained for this revision was clearly positive and raised only minor points requiring clarification. My own re-evaluation confirms that the manuscript is technically sound, adheres to PLOS ONE’s data and methodological transparency requirements, and has been substantially improved in response to both the reviewer’s comments and the initial editorial screening. The remaining issues concern only brief clarifications regarding material classification and sample provenance, which are required for acceptance.

We look forward to receiving your revised manuscript.

Kind regards,

Przemysław Mroczek, Dr. hab.

Academic Editor

PLOS ONE

Journal Requirements:

Additional Editor Comments:

Dear Authors,

Thank you very much for submitting the revised version of your manuscript and for your careful and constructive responses to the comments raised during the evaluation process. One external review was obtained for this revision, and the reviewer was clearly positive regarding the scientific soundness, methodological transparency, data availability and clarity of presentation. The reviewer recommended only minor revisions and highlighted that the manuscript is well structured, innovative in its application of CNN-based segmentation to micromorphological materials, and valuable for improving reproducibility in archaeological micromorphology. I appreciate the considerable effort you have made to address both the reviewer’s suggestions and the earlier editorial requirements.

I have re-evaluated the revised manuscript in full, including the supplementary materials and the publicly accessible code and dataset repositories, and I also conducted an internal check of consistency, terminology and compliance with PLOS ONE policy. The manuscript has improved substantially, and in its present form it is close to being suitable for acceptance. Before proceeding, however, I kindly request that you address two minor issues in a final revision. First, although you explain why flint and obsidian were merged into a single class, it would be helpful to state explicitly whether the model performs equally well on both materials and whether this merging was driven exclusively by sample size or also by their practical indistinguishability in PPL/XPL imagery. This brief clarification, which the reviewer also noted, will be useful for future users of the dataset. Second, the manuscript currently describes the thin sections as originating from “different contexts, regions and time periods”. For transparency and archaeological replicability, please provide one additional clarifying sentence indicating at least the broad geographic areas or approximate chronological ranges represented in the dataset. This does not require detailed provenance information, merely a minimal contextual indication.

I look forward to receiving your revised version.

With best regards,

Przemysław Mroczek

Academic Editor

Reviewers' comments:

Reviewer's Responses to Questions

**Comments to the Author**

1. If the authors have adequately addressed your comments raised in a previous round of review and you feel that this manuscript is now acceptable for publication, you may indicate that here to bypass the “Comments to the Author” section, enter your conflict of interest statement in the “Confidential to Editor” section, and submit your "Accept" recommendation.

Reviewer #1: (No Response)

2. Is the manuscript technically sound, and do the data support the conclusions?

Reviewer #1: Yes

3. Has the statistical analysis been performed appropriately and rigorously?

Reviewer #1: Yes

4. Have the authors made all data underlying the findings in their manuscript fully available?

Reviewer #1: Yes

5. Is the manuscript presented in an intelligible fashion and written in standard English?

Reviewer #1: Yes

6. Review Comments to the Author

Reviewer #1: The manuscript “Improving micromorphological analysis with CNN-based segmentation of flint/obsidian, bone and charcoal” presents an innovative application of CNNs to archaeological micromorphology. Overall, the study demonstrates the potential of deep learning for improving reproducibility and consistency in archaeological micromorphology, while also acknowledging dataset limitations, class imbalance, and annotation challenges. The work is clearly structured, the dataset and methods are transparent, and the results are promising. I appreciate the effort to make code and data openly available. However, there are aspects that would benefit from clarification and reorganisation, mainly concerning consistency in terminology, the structure of the introduction and results, and the archaeological contextualisation.

Major comments

Abstract

The phrase “(…) flint (including obsidian)” suggests that obsidian is a subtype of flint. Please rephrase to clarify that they were clustered together for methodological reasons, not because of material taxonomy. Correct it throughout.

Moreover, are flint and obsidian truly indistinguishable in this approach? Is the model equally reliable with both, or could there be differences? Since not all Palaeolithic contexts contain obsidian, and many use other types of chert, this limitation should be acknowledged.

Introduction

Lines 14-17: add references.

Lines 22-23: explain the statement and cite if not your own reasoning.

Lines 24-25: this does not need a new paragraph.

Lines 35-51: this is the core of the introduction, but it is currently diffuse. Please rewrite to highlight clearly the aim, how, and significance of the study.

Lines 44-48: add references.

Lines 52-58: the tone is too didactic; I suggest trimming or rephrasing. The last paragraph of the introduction should state succinctly: what you propose, why it matters, and what novelty it brings to the field.

Related work

As written, this section feels isolated. Consider integrating the discussion of ML applications into Introduction, Methods, or Discussion instead of leaving it standalone.

Lines 66-74 (Big Data vs Slow Data): this paragraph is disconnected. If you keep it, please explain why this distinction matters. Line 71 needs a reference.

Materials and methods

In Fig. 1 use the same terminology (Flint/Obsidian) as in the text.

Table 2: unify the number format and apply a consistent rounding rule across rows.

Line 183: no need to re-explain the acronym after the first mention; please correct throughout.

Please provide more detail about the origin of contexts and the experimental samples: where do they come from, and how were they created?

Results

Table 6: write Flint/obsidian or explain the abbreviation.

This section contains methodological notes and interpretive comments that should be moved to Methods and/or Discussion. The Results section should be reserved for data only.

Lines 273-275: this partly duplicates the caption of Fig. 5. Please integrate or remove repetition.

Discussion

Lines 339-341: the phrasing is too promotional; please reword to avoid sensational tone.

The archaeological impact would benefit from more concrete examples: how exactly would this tool improve interpretation of hearth features, stratigraphic complexity, or activity areas?

The limitations section should note that charcoal, in particular, requires more samples and more morphological variability to improve training.

Other comments

In figure titles, you can avoid phrasing such as “Results of …”.

Avoid one-sentence paragraphs throughout the text; merge with adjacent ones for smoother reading.

A workflow figure (dataset → annotation → model training → results) would improve accessibility.

Standardise terminology: Flint/Obsidian throughout.

Ensure all abbreviations are expanded at first use (IoU, PPL, XPL) and no need to expand them in the following occurrences (e.g., line 183).

Improve figure/table captions so they are more self-explanatory.

7. PLOS authors have the option to publish the peer review history of their article (what does this mean?). If published, this will include your full peer review and any attached files.

Reviewer #1: No

---

## [Author Response · Author response to Decision Letter 2]

18 Dec 2025

The authors greatly appreciate the provided suggestions, which have been addressed as described below:

- The phrase "(...) flint (including obsidian)" suggests that obsidian is a subtype of flint. Please rephrase to clarify that they were clustered together for methodological reasons, not because of material taxonomy. Correct it throughout.}

The text has been rephrased to "lithic fine-grained debitage (flint and obsidian)" and corrected throughout the whole manuscript.

- Moreover, are flint and obsidian truly indistinguishable in this approach? Is the model equally reliable with both, or could there be differences? Since not all Palaeolithic contexts contain obsidian, and many use other types of chert, this limitation should be acknowledged.

The following sentence has been included in "Annotating difficulties and future directions" and "Conclusions" respectively: "the dataset will be expanded to include a greater number of these materials, along with additional fine-grained lithic raw materials such as chert and quartzite," and "Furthermore, several archaeological contexts use chert, quartzite or other fine and coarse-grained materials that are not represented in this study."

= Introduction:

- Lines 14--17: add references.

References Angelucci (2003) and Huisman et al. (2014) were added to the paragraph.

- Lines 22--23: explain the statement and cite if not your own reasoning.

- Lines 24--25: this does not need a new paragraph.}

The statement was merged with the next paragraph, where it is explained.

- Lines 35-51: this is the core of the introduction, but it is currently diffuse. Please rewrite to highlight clearly the aim, how, and significance of the study.

To enhance the clarity of the aim and objectives of the article, we propose to merge the two paragraphs into one: "To enhance the reliability of micromorphological data, it is essential to maximize objectivity and, where possible, introduce quantitative precision to support interpretation. With this in mind, the primary aim of our project is to develop a tool using image--based ML that facilitates the identification and quantification of some of the most common materials in Palaeolithic contexts and thin sections: bone (fresh, degraded, burnt and calcined), charcoal and fine-grained lithic debitage (flint and obsidian)."

- Lines 44-48: add references.

As suggested in the response above, these paragraphs have been removed.

- Lines 52-58: the tone is too didactic; I suggest trimming or rephrasing. The last paragraph of the introduction should state succinctly: what you propose, why it matters, and what novelty it brings to the field.

The paragraph has been removed. The introduction now finalizes with stating the aims of the article.

= Related work

- As written, this section feels isolated. Consider integrating the discussion of ML applications into Introduction, Methods, or Discussion instead of leaving it standalone.

The paragraph has been moved to the introduction, before the "aims" paragraph.

- Lines 66-74 (Big Data vs Slow Data): this paragraph is disconnected. If you keep it, please explain why this distinction matters. Line 71 needs a reference.

The paragraph "Big Data vs Slow Data" has been removed, as well as lines 86-93 as they felt disconnected.

= Materials and methods

- In Fig. 1 use the same terminology (Flint/Obsidian) as in the text.

The legend of figure 1 has been modified to match the terminology of the rest of the work.

- Table 2: unify the number format and apply a consistent rounding rule across rows.

All numeric data in table 2 have been unified to fixed format with two decimal digits.

- Line 183: no need to re-explain the acronym after the first mention; please correct throughout.

All redefinitions of the IoU acronym have been removed, except the first instance and within the abstract.

- Please provide more detail about the origin of contexts and the experimental samples: where do they come from, and how were they created?

The following sentence has been included for contextual indication, following the editor's comments: "These photomicrographs were obtained from [...] contexts where these materials are abundant. Chronologies and geographical areas include European and Near Eastern Middle Palaeolithic, Iberian Peninsula Neolithic and Ancient Canarian and experimental samples studied in Mallol et al. (2013)."

= Results

- Table 6: write Flint/obsidian or explain the abbreviation.

An explanation of the abbreviation has been included in the caption of Table 6.

- This section contains methodological notes and interpretive comments that should be moved to Methods and/or Discussion. The Results section should be reserved for data only.

The second paragraph of the Results section has been partially removed, as its information was already discussed in the "Materials and Methods" section. Similarly, several paragraphs discussing the meaning of the results have been moved to the beginning of the Discussion section.

- Lines 273-275: this partly duplicates the caption of Fig. 5. Please integrate or remove repetition.

The redundant paragraph has been removed in favor of the figure caption.

= Discussion

- Lines 339-341: the phrasing is too promotional; please reword to avoid sensational tone.

The sentence has been reworded as "The possibility of automating part of the identification and quantification process constitutes a practical improvement: it would reduce the time micromorphologists spend on manual annotation and counting, while still providing results that are consistent and dependable."

- The archaeological impact would benefit from more concrete examples: how exactly would this tool improve interpretation of hearth features, stratigraphic complexity, or activity areas?

The following paragraph was added: "This methodology would enhance the quantification of these elements in archaeological features. For instance, in the characterization of combustion structures, where charcoals are frequently found, or the identification of surface occupations and activity areas, characterized by high abundance of bones and fine-grained lithic debitage."

- The limitations section should note that charcoal, in particular, requires more samples and more morphological variability to improve training.

The paragraph was rephrased as follows: "At the same time, the dataset will be expanded to include a greater number of these materials, along with additional fine-grained lithic raw materials such as chert and quartzite, as well as charcoal exhibiting greater morphological variability and bone fragments. Other materials expected to be incorporated into the dataset include pottery fragments, specific rock lithologies, and malacofauna, thereby broadening the applicability of the model."

= Other comments

- In figure titles, you can avoid phrasing such as "Results of ...".

The captions of figures 2, 3 and 4 have been simplified by removing the redundant wording.

- Avoid one--sentence paragraphs throughout the text; merge with adjacent ones for smoother reading.

To improve readability, all one-sentence paragraphs have been merged into larger bodies, as marked by their initial letters in the highlighted manuscript, or removed.

- A workflow figure (dataset → annotation → model training → results) would improve accessibility.

A workflow figure has been added at the end of the Materials and Methods section, as suggested.

- Standardise terminology: Flint/Obsidian throughout.

All occurences of the term have been standardised in the manuscript.

- Ensure all abbreviations are expanded at first use (IoU, PPL, XPL) and no need to expand them in the following occurrences (e.g., line 183).

All abbreviations have been fixed, expanding them only on their first appearance.

- Improve figure/table captions so they are more self-explanatory.

The captions of tables 6 and 7 have been reworded to be more self-explanatory.

---

## [Editor Report · Decision Letter 2]

21 Dec 2025

Improving micromorphological analysis with CNN-based segmentation of flint/obsidian, bone and charcoal

PONE-D-25-43021R2

Dear Dr. Hernández-Aceituno,

We’re pleased to inform you that your manuscript has been judged scientifically suitable for publication and will be formally accepted for publication once it meets all outstanding technical requirements.

Kind regards,

Przemysław Mroczek, Dr. hab.

Academic Editor

PLOS One

Additional Editor Comments (optional):

Dear Dr. Hernández-Aceituno,

thank you for submitting the revised version of your manuscript, “Improving micromorphological analysis with CNN-based segmentation of flint/obsidian, bone and charcoal” (PONE-D-25-43021R2), to PLOS ONE.

I have now completed my editorial assessment of the revised manuscript and the authors’ responses to the previous comments. I am pleased to inform you that your manuscript is accepted for publication in PLOS ONE.

The revised version adequately and clearly addresses all substantive points raised during the review process. In particular, the manuscript now demonstrates improved focus and structure, a more balanced and appropriate tone in the Discussion, clearer articulation of methodological choices and limitations, and full compliance with PLOS ONE’s policies on data availability, transparency, and reporting. The study meets the journal’s criteria for methodological soundness and reproducibility, and it represents a valuable contribution to the development of quantitative and computational approaches in archaeological micromorphology.

Before the manuscript proceeds to production, please note the following minor editorial observations, which are offered for your information only and do not require a revised submission or formal response:

-The Conclusion section contains some thematic overlap between consecutive paragraphs; light streamlining during copyediting may further improve clarity.

-Where applicable, copyediting may slightly harmonise phrasing related to model limitations and generalisability, without altering the substance of your conclusions.

These points will be handled, as appropriate, during the production and copyediting stages.

I thank you for submitting your work to the journal and for your careful and constructive engagement throughout the review process. I look forward to seeing your article published.

Kind regards,

Przemysław Mroczek, Dr hab.

Academic Editor
---

## [Editor Report · Acceptance letter]

PONE-D-25-43021R2

PLOS One

Dear Dr. Hernández-Aceituno,

I'm pleased to inform you that your manuscript has been deemed suitable for publication in PLOS One. Congratulations! Your manuscript is now being handed over to our production team.

Kind regards,

on behalf of

Dr. hab. Przemysław Mroczek

Academic Editor

PLOS One